# Emotional Temperament and Character Dimensions and State Anger as Predictors of Preference for Rap Music in Italian Population

**DOI:** 10.3390/ijerph192013650

**Published:** 2022-10-21

**Authors:** Carmenrita Infortuna, Fortunato Battaglia, David Freedberg, Carmela Mento, Fiammetta Iannuzzo, Rosa De Stefano, Clara Lombardo, Maria Rosaria Anna Muscatello, Antonio Bruno

**Affiliations:** 1Psychiatric Unit, University Hospital “G. Martino” of Messina, 98125 Messina, Italy; 2Department of Medical Sciences and Neurology, Hackensack Meridian School of Medicine, Nutley, NJ 07110, USA; 3Department of Art History and Archaeology and Italian Academy for Advanced Studies in America, Columbia University, New York, NY 10027, USA; 4Department of Biomedical and Dental Sciences, Morphological and Functional Images, University of Messina, 98121 Messina, Italy

**Keywords:** emotional temperaments, trait anger, music preference, psychological determinants, cross-sectional

## Abstract

The objective of the current work was to examine the relationships between affective temperaments dimensions, trait anger, and the preference for rap music in a sample of Italian adults. An online survey was administered to 662 subjects. We assessed preference for rap music using a Likert scale. Ina addition, we investigated the sample’ affective temperament traits using the Memphis, Pisa, Paris, and San Diego (TEMPS-A) short scale and the trait anger by using the Trait Anger component of the State Trait Anger Expression Inventory 2 (STAXI-2). Multiple linear regression indicated that cyclothymic temperament score, hyperthymic temperament score, and trait anger scores were positive predictors while older age, and depressive temperament and higher education score were negative predictors of preference for rap music. The results expand previous literature on personality and music preference indicating the association of high energy/high activity temperaments and trait anger to preference for rap music.

## 1. Introduction

Taste for artistic expressions represents an important factor that defines the way we relate to people and how we interpret their thoughts [1]. In recent years, live streaming, social media, and the internet have made music available and relatively inexpensive, leading to an increase in the popularity of this artistic genre [2]. There is a strong link between our emotions and the music that we like. Furthermore, music preferences define and drive social interaction and represent a relevant factor influencing young people topic of discussion [1] and bonding with their peers [3]. Several theoretic approaches have been developed to explain the psychological factors associated with music preference. For instance, previous research indicates that age, gender, and personality are important determinants for music preference. Studies indicate that adolescence is the time of life when we develop our musical preference, which shows developmental changes throughout the lifespan [4]. A taste for more energetic music seems more often associated with the male gender and a younger age [5]. The bulk of research on psychological determinants for music preference focused on cognitive styles and personality dimensions. Personality refers to individual differences in characteristic patterns of thinking and feeling so that people with specific personality traits exhibit similar behaviors when exposed to analogous situations across time. Previous studies described several theoretical approaches explaining the association between personality traits and musical preferences. The *uses-and-gratifications theory* implies that we select a specific music genre to fulfill our psychological needs and expectations at that moment [6]. According to the *theory of optimal arousal,* it was demonstrated that the preference for more arousing types of music would be associated with lower resting arousal and higher sensation seeking personality [7]. It is worth noting that, within personality research and music, results are sometimes heterogeneous and inconclusive. A number of studies have explored the association between the Big Five personality dimensions and the five factors musical styles (MUSIC model) [8]. According to this framework, which is consistent with interactionist theories [9], it was suggested that personality traits can predict music preference, reflecting a psychological predisposition for specific music genres [8,10,11,12,13]. For instance, extraverts tend to select rhythmic and high-energy contemporary genres such as rap, electronica, Latin, and Euro-pop genres. This association seems to be universal, across countries and cultures [14]. Despite the aforementioned results, a recent metanalysis has challenged the predictive value of personality dimensions (small correlation coefficients), and highlighted the need to explore different psychological characteristics to explain differences in musical style preferences [11].

Of a particular interest are the psychological determinants of preference for music genres with remarkable social significance as rap music. Hip-hop/rap is a music style developed in the 1970s in the South Bronx as a “street” cultural and artistic movement characterized by political advocacy and the need for fighting systemic injustices that plagued these communities. In recent decades, rap became a commercialized genre that gained worldwide popularity. Empirical reports have indicated that rap has been associated with socially inappropriate behaviors such as aggression, violence, homophobia, misogynistic beliefs, and drug use [15]. To date, it remains to be established whether rap music lyrics, subculture, and lifestyle are capable of negatively influencing behaviors [16] or if negative behaviors are to be linked to individual psychological traits. For instance, to date no research has explored the role of innate, individualized levels of reactivity upon which personality is later built on rap music preference. Temperaments are thought to represent the heritable and neurochemically determined emotional reactivity to stimuli. The five temperaments (depressive, hyperthymic, cyclothymic, anxious, and irritable) assessed with the short version the TEMPS-A Scale (Temperament Evaluation of Memphis, Pisa, Paris and San Diego–Autoquestionnaire) are stable through the lifespan, and they unfold into personality characteristics during development [17]. Previous works have demonstrated that temperament traits are associated with health risk behaviors [18,19], esthetic judgment, and preference for specific film genres [20]. A further psychological dimension which appears understudied within the literature of music preference is trait anger, which is the individual predisposition of experiencing more frequent, severe, and intense anger associated with less thoughts of self-control [21]. A body of experimental studies suggests that temperaments characterized by a negative emotionality appear linked to anger and that this link underlies to approach-related affects, such as excitement [22]. In line with this idea, trait anger might unfold into a rebellious reactivity to an unfair violation of what the person counts upon [23] that may be depicted in high-energy lyrics. Despite the aforementioned literature, it remains to be determined whether these basic individual psychological characteristics that control emotion regulation processes and reactivity are reflected in rap music preference. Furthermore, rap music is now the most popular music genre In Italy and its popularity seems to cross cultural and social backgrounds. Hence, the investigation of the psychological determinants of preference for rap music in different cultures should consider instruments that address innate traits, and emotional reactivity that drives individual taste for certain musical characteristics and behaviors. Previous studies indicates that individual temperaments are an important factor for the development of aggression in children [24]. Thus, the innate emotional reactivity could serve as a foundation linking a taste for a specific music with sensation seeking, health-risk behavior, and aggression across cultures.

The aim of our present cross-sectional study was to investigate the association between demographic variables (age, gender), affective temperaments traits, and trait anger on the individual’s choice of listening to rap music in a sample of Italian adults. In keeping with prior theoretical models and past research, we hypothesized that age would predict preference for rap music. Furthermore, we hypothesized that stable, inherited traits which determine the basic levels of reactivity, mood, and energy are associated with taste for rap music and associated behaviors.

## 2. Methods

A data collection questionnaire was developed to investigate the factors associated with a preference for rap music. This study was undertaken in Italy (October 2020–February 2021). This study employed an anonymized, freely available online survey distributed on social media (websites, Facebook, and Twitter), and data were collected via Google Forms. Only completed forms were stored and analyzed. Adults aged ≥18 years and ≤55 were eligible to participate. All respondents were provided with information about the aims, methods and expected benefits of the study. All data were collected anonymously, there was no incentive to participate in the study, and the participants could withdraw at any time. The submission of the survey was considered informed voluntary consent. The study was approved by the Institutional Review Board of the University of Messina, Italy.

### 2.1. Measures

Participants indicated their preference for rap music using a 5-Likert scale: “In general, how much do you like listening to the rap music: (1) really dislike; (2) dislike; (3) neither like nor dislike; (4) like; (5) really like.

Affective temperaments were assessed using the TEMPS-A scale, short version. The scale yields four affective temperament dimensions: cyclothymic, depressive, irritable, hyperthymic, and anxious. It is a self-report, yes-or-no type questionnaire *(My ability to think varies greatly from sharp to dull for no apparent reason. Yes; No)*. Each item in The Italian version of the instrument was previously validated and used in an online survey with Italian participants [19,25]. We assessed individual’s disposition to become angry by using the Trait Anger component of the State Trait Anger Expression Inventory 2 (STAXI-2) [21]. Each item is measured on a Likert ranging from 1 to 4 *(I argue with others. 1 = Not At All; 2 = Somewhat; 3 = Moderately So; 4 = Very Much So).* The Trait Anger component comprises 10 items assessing how often angry feelings are experienced over time and has two subscales (angry temperament and reaction anger). The Italian version of the instrument was previously validated [26].

### 2.2. Statistical Analysis

Only completed questionnaires were analyzed. Data were analyzed with descriptive statistics (numbers and percentage). Furthermore, predictors (demographic and psychological variables) were assessed using a multiple linear regression. The data were analyzed with IBM SPSS Statistics 23.0 (Armonk, NY, USA: IBM Corp). A *p* value < 0.05 was considered statistically significant,

## 3. Results

The study sample characteristics are shown in Table 1. Participants were on average 28 years old (SD = ±9.27) with a range of 18–55 years old. There were 662 participants: 216 male (32.6%%) and 446 female (66.4%). The results indicated that 15.1% liked rap music and 7.1% really liked rap music.

Temperament traits scores and trait anger scores are reported in Table 2.

All measures showed an acceptable reliability, assessed with the Cronbach’s alpha level (cyclothymic trait score = 0.702; depressive trait score = 0.76; irritable trait score= 0.81; hyperthymic trait score = 0.75; anxious trait score 0.720; Trait Anger score score = 0.76).

Multiple linear regression was used to test if demographic variables (age, gender, education), temperament traits, and state anger significantly predicted the preference for rap music.

The overall regression was statistically significant (R2 = 10.7, F (10,662) = 6.9, *p* = 0.001). The results indicated that male cyclothymic temperament score (β = 0.11, *p* = 0.02), hyperthymic temperament score (β = 0.09, *p* = 0.03) and Trait anger (β = 0.09, *p* = 0.04) were positive predictors; on the contrary, older age (β = −0.21, *p* = 0.001), higher education (β = −0.10, *p* = 0.01) and depressive temperament score (β = −0.09, *p* = 0.04) are negatively associated with higher preference for rap music (Table 3).

## 4. Discussion

This is the first study describing relationships between affective temperamental traits, trait anger, and preference for rap music in an Italian adult population. The results indicate that high-energy temperaments (cyclothymic and hyperthymic), and Trait anger score predicts higher likelihood for enjoying rap music. On the contrary, low-energy temperament (depressive), older age, and higher education are negative predictors.

Concerning the demographic variables, the results are keeping with previous studies indicating the association of younger age and lower education with the preference for energetic and rhythmic music like rap, assessed by using the MUSIC model [10]. One possible explanation is consistent with the socialization hypothesis, based on the Social Learning Theory [27]: rap artists are among the most popular celebrities and teenagers considered them role models, regularly watch music videos, and stay up-to-date on music artists’ lifestyle [28]. Hence, considering an ecological framework [29], rap might be considered like an exosystem. Youngsters are surrounded by media content, imitating what is described in the lyrics and depicted in music videos. Microsystems like family and friends may heighten or minimize the influence of negative messages and the consequent behaviors (drug use, aggression) [28,30]. Furthermore, evidence indicates that music preference develops in childhood, has biological basis (hormones, reward system), and reflects the internal characteristics of the listener [31], The association between temperament traits and the preference for rap music is consistent with this hypothesis. In our study, cyclothymic traits were associated with higher preference for rap music. This temperament trait is linked to unpredictable mood instability and the tendency of quick fluctuations in energy levels, impulsiveness, anger, and harm avoidant behaviors [32]. Furthermore, previous studies showed that cyclothymic traits are associated with smoking, obesity, and recreational cannabis use [25,33]. Thus, cyclothymic temperament traits may contribute to the development of risky and aggressive behaviors in individuals who prefer rap music [34]. Future longitudinal studies are needed to address this causal link in rap music followers. Our results indicate that hyperthymic traits are positive predictors of preference for rap music. These traits are characterized by optimistic view, high self-esteem, high activity/energy level, novelty seeking behaviors. Since extraversion and hyperthymic temperament traits are positively correlated [35], our results are in keeping with previous studies indicating higher preference for rap music in extroverts [10]. On the contrary, we found that depressive temperaments, which are characterized by low-energy and activity are negatively associated with the preference for rap music. Since temperaments are individual differences in the level of reactivity, mood, and energy already evident in infants prior to the development of personality, our results expand the literature on psychological determinants of music preference [10]. This is of relevance since the correlation coefficients between extraversion and preference for rhythmic music (which included rap) previously reported are rather low [11] indicating the need for studies investigating different psychological determinants. It is likely that individual cognitive styles and mood affect the processing of the emotions represented in music (anger, fear, happiness, sadness, and tenderness) and may play a pivotal role in music preference [36,37]. Thus, future studies should consider trait empathy as a relevant variable in the investigation of the psychological factors associated with preference for rap music [38].

We demonstrated that the trait anger is associated with preference for rap music. Trait anger cand be considered as the individual difference in feeling anger as an emotional state. Previous research showed that trait anger may be associated with aggressive behavior [39], troubled interpersonal relationships, increased social conflict, and decreased social support [40]. Hence, high traits anger and associated emotional arousal highlight a possible foundation leading to future psychosocial vulnerability in consumers of rhythmic, high-energy music.

The study has limitations. First, the cross-sectional/correlational design cannot evaluate causality nor ensure generalizability to different populations. Online surveys may be biased because of the participants’ interest in the topic, computers and social media skills, and access to the internet. Furthermore, the role of more socioeconomical, cultural, and environmental variables need to be addressed in future research. It is likely that the geographical and sociocultural context would be of pivotal importance and would affect the preference for an energetic music genre. Furthermore, since this is the first study investigating temperament traits and music preference, the results should be taken with caution and confirmed in future, larger studies. In addition, the instruments we have used in our studies might not be optimal for addressing our research questions. Future studies should include different instruments (which do not rely on self-assessment), a different study design, and a different statistical analysis which can better address the interplay between different variables and the role of latent variables.

## 5. Conclusions

In this study, we described the predictors of preference for rap music in an Italian sample of adults. Our results indicate that, after controlling for demographic variables, cyclothymic and hyperthymic temperament score and trait anger score were positive predictors while depressive temperament score are negative predictors. Our novel findings expand previous literature indicating the contribution of innate, stable traits to music preference.

## Figures and Tables

**Table 1 ijerph-19-13650-t001:** Demographic characteristics of the participants (*n* = 662).

SAMPLE N = 662		
**Age** (mean and st.deviation)	28.2 ± 9.2	
**Gender** (Frequency and Valid Percent)		
M	216	32.6%
F	446	66.4%
**Education** (Frequency and Valid Percent)		
Elementary	2	0.3%
Secondary school	33	6.5%
High school	290	43.8%
University and Post-degree	327	49.4%

**Table 2 ijerph-19-13650-t002:** Descriptive statistics of the psychological study variables (mean ± SD).

Temperament Dimensions (TEMPS-A, Short Version)	
Cyclothymic	5 ± 3.12
Depressive	2.5 ± 2.04
Irritable	1.4 ± 1.54
Hyperthymic	4.3 ± 1.9
Anxious	1.4 ± 1.08
**STAXI-2**	
Trait anger	16.9 ± 3.5

**Table 3 ijerph-19-13650-t003:** Multiple regression analyses with the study’s predictors (age, gender, education temperament traits and state anger) of preference for rap music.

Effect	Estimate	SE	95% CI		*p*
			LL	UL	
(Constant)		0.36	1.88	3.29	0
GENDER	0.07	0.09	0.00	0.34	0.06
AGE	−0.21	0.01	−0.03	−0.02	<0.001 **
EDUCATION	−0.10	0.07	−0.30	−0.04	0.01 **
Cyclothymic	0.11	0.02	0.01	0.07	0.02 *
Depressive	−0.09	0.02	−0.10	0.00	0.04 *
Irritable	−0.03	0.03	−0.08	0.05	0.58
Hyperthymic	0.09	0.02	0.01	0.09	0.03 *
Anxious	0.00	0.04	−0.08	0.08	0.93 *
Trait anger	0.09	0.01	0.00	0.05	0.04

* *p* < 0.05; ** *p* < 0.01.

## Data Availability

The data will be available upon request to the principal investigation.

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
