# Peer review of "Emotional Temperament and Character Dimensions and State Anger as Predictors of Preference for Rap Music in Italian Population"

_ijerph, 2022, doi:10.3390/ijerph192013650_

Round 1

Reviewer 1 Report

The introduction too briefly and superficially describes previous work on personality traits and music preference. It for instance misses to reports on/discuss a large number of substantial works performed on the matter (e.g., work of Vuoskovski, Eerola, …). Moreover, there is much debate on the topic (there is a large individual variability, often being more crucial than certain personality traits + often the personality traits relate to other items such as demographics, nature/nurture aspects etc.), a much more critical account should be provided, also discussing more general items such as correlations/causation regarding the matter. Also, the researchers should explain more clearly why the hypothesis put forward makes sense as well as why it is relevant. Put more generally: going over the introduction, it seemed too swiftly written, lacking a thorough description of the back matter as well as missing a critical perspective on the topic.

The STAXI-2 Likert scale went from 1-4, which does not include a neutral option. Why were participants not provided with a middle option? Now, they were pushed to provide an answer in a certain direction, while they might not have had an opinion or idea (self-assessment should never be ‘pushed’, often people don’t have a clue, which is normal and which is fine; pushed answers are questionable, this should be taken into account in the study design and/or discussion).

Is self-assessment a good way to test anger traits? This can be strongly debated and should be discussed. Moreover, males and females, different age groups, etc. might be inclined to report differently on this trait, depending on social expectancies for instance. 

Similar as for the introduction, the discussion misses depth and critical notes, embedded in the narrative (rather than quickly mentioning some limitations at the end of the discussion). The findings should be linked and embedded more strongly with/in previous work and theory. Moreover, results should be discussed more critically, in a thorough way (to give an example; what does the link with age mean? And could it interact with other aspects, for instance because age is commonly related to specific traits/features/…). As well as for the introduction, the discussion requires thorough rewriting in order to be significantly firmer and more comprehensive.

What was the response rate?

Reporting “p=<.000” does not make sense (it should be .001 or .0001; it can’t go below 0).

The tables are sometimes a bit sloppy. For instance, in table 1 above each column, it should be put what the numbers refer to (percentages, n’s, …).

There are several typos and language errors/oddities, please go over the text thoroughly again to correct them. E.g., page 1 ‘people’ topics’ (the s got lost); page 3 ‘The participants were 662, 446 (67.4%) female and 216 (32.6%)’ (the word ‘male’ got lost and the sentence is oddly constructed), and so on.

There are inconsistencies in reporting numbers after the comma (currently ranging from 1 to 3).

Author Response

We thank the Reviewer for the comments. In this revised version of the manuscript, we have included all the suggestions.

  • …The introduction too briefly and superficially describes previous work on personality traits ………….. critical account …… correlations. Also, the researchers should explain more clearly why the hypotheses … critical perspective on the topic.

We agree with the Reviewer’s comments. We did a major revision and included 14 new references (the suggested references were included in the Discussion). We now have a more in-depth description of the proposed theories underlying music preference and a description of the relevant literature highlighting the association of demographic variables (age , gender) and psychological determinates (Big five, the MUSIC model) with music preference. Furthermore, as suggested, we are now reporting the limitation of previous studies. The weak correlation of the personality dimensions with a taste for rhythmic music which include Rap music prompted us to investigate different determinants. We believe that innate emotional relativity plays a role in determining music preference. We are now suggesting to study innate temperament traits (emotional temperament and trait anger) which may be associated with preference for rap music and may explain previously described behaviors (aggression, violence, sensation-seeking, drug use…).  If innate emotional reactivity (addressed with the study of temperament traits) is a relevant factor, we might find an association in a population from a Country which does not share the social and cultural background that characterized Rap music since its development in the 70s. For this reason, we are performing a study in an Italian population. Rap is the most popular genre in Italy and it was often associated to maladaptive behaviors. An association with temperament traits would suggest a pivotal role for specific innate emotional characteristics that could lead to preference for music with specific styles (rhythm, lyrics,,,)  and at the same time , predispose to maladaptive behaviors. 

The gap in the literature, research questions , and hypotheses are know stated .

  1. The STAXI-2 Likert scale….

We agree with the Reviewer. Both scales have some limitation. We have used these instruments because of the large body of literature and because both scales were used before in online cross-sectional studies. We are including comment in the study Limitations. Furthermore, since this is the first study investigating temperament traits and preference for Rap music our results will need to be futher confirmed, in larger studies, using different variables, and a different study design

  1. Is self-assessment a good way to test anger traits?

We have included this comment among the study limitations

  1. ...Discussion

As suggested by the Reviewer, we revised the discussion. We now have a more organized description of the literature, the relevant theories. We expanded the Limitation sections . Since the study is cross-sectional,  we kept the Discussion short and schematic and removed overstatements. Furthermore, this is an online surgery and the survey link was posted on internet on Italian websites, discussion group, and consumers groups . For this reason, we cannot report the response rate.

  1. Reporting “p=<.000”..

We have corrected the typo

  1. Table 1,3

We thank the Reviewer for pointing to our attention the typos.  The description is reported in parenthesis (mean and st. deviation, mean and valid percent). We revised Table 3 according to the Reviewer’s suggestion.

  1. Typos

 We corrected all the typos pointed to our attention by the Reviewer. In case of acceptance, we will follow up with the editorial office and, if needed, we will arrange for a revision from a native English speaker.

Reviewer 2 Report

Title: include information about sample

Keywords: do not repeat from those who already are in title

Introduction: provide more information about similar previous empirical research.

Measures: describe each factor, writing one item as example.

Discussion: improve the comparison of your results with other previous results, not only remarking if they are in line or not but also providing potential explanations based on up-dated references. Increase the limitations and the future works that could be developed.

References: Review journal's format guidelines.

Author Response

We thank the Reviewer for the comments. Please find included the revised manuscript which include the Reviewer’s suggestions.

  1. Title: include information about sample

- Information about sample was added.

  1. Keywords: do not repeat from those who already are in title

- We revised the keywords

  1. Introduction: provide more information about similar previous empirical research.

The did a major revision of the Introduction We have included 14 new referenced and discussed previous literature (demographic, cognitive, and psychological determinants; theoretical models, MUSIC model). We have highlighted the limitation of previous research on personality. Furthermore, we revised the research questions and hypothesis.

  1. Measures: describe each factor, writing one item as example.

We have included a sample item and the description for both scales

  1. Discussion: improve the comparison of your results with other previous results, not only remarking if they are in line or not but also providing potential explanations based on up-dated references. Increase the limitations and the future works that could be developed.

We have revised the Discussion according to the Reviewer’s suggestions

  1. References: Review journal's format guidelines.

In case of acceptance, we will follow up with the editorial office (formatting, English editing…)

Round 2

Reviewer 2 Report

Dear authors,

this new version is, in my humble view, much better than the previous one.

I congratulate you for your effort.

Author Response

Thank you very much for the kind comments.